# NLMs: Augmenting Negation in Language Models

**Rituraj Singh, Rahul Kumar** and **Vivek Sridhar**
Samsung R&D Institute India, Bangalore
{rituraj.s, rahul.k4, v.sridhar}@samsung.com

## Abstract

Negation is the fundamental component in a natural language that reverses the semantic meaning of a sentence. It plays an extremely important role across a wide range of applications, yet they are under-represented in pre-trained language models (LMs), resulting often in wrong inferences. In this work, we try to improve the underlying understanding of the negation in the pre-trained LMs. To augment negation understanding, we propose a language model objective with a weighted cross-entropy loss and elastic weight consolidation regularization. For negated augmented models, we reduce the mean top 1 error rate for BERT-base to l.1%, BERT-large to 0.78%, RoBERTa-base to 3.74%, RoBERTa-large to 0.01% on the negated LAMA dataset that outperform the existing negation models. It minimizes the mean error rate by a margin of 8% and 6% for original BERT and RoBERTa models. We also provide empirical evidences that negated augmented models outperforms the classical models on original as well as negation benchmarks on natural language inference tasks.

## 1  Introduction

Negation plays a pivotal role in many natural language understanding tasks, such as sentiment analysis, question answering, Natural Language Inference (NLI) tasks, etc. While large language models such as BERT have pushed state-of-the-art and are being used widely across various domains, it fails dramatically to understand negation.

Negation relates an expression $e$ to another expression with a meaning that is in some way contrary to the meaning of the original sentence $e$. Negation is ubiquitous in English language text and comprises approximately 25% sentences with negation clues or genre in some form (Hossain et al., 2020). Despite this fact, understanding of negation is under-represented in the language models. The current Language Models (LMs) cannot distinguish between negated and non-negated forms of

masking tasks (Kassner and Schütze, 2020). For example, when asked to predict the [MASK] token in the sentences: The capital of South Korea is [MASK]. and The capital of South Korea is not [MASK].; BERT often generates the same answer as "Seoul". The exercise gives evidence that LMs do not appropriately model the distribution of negation sentences. LMs when fine-tuned on NLI tasks, current pre-trained models tend to misclassify the examples which contain negations as contradictions when the true label is neutral or entailment.

In this work, we address the challenge of augmenting negation understanding of language models and alleviating the model bias. To enhance negation understanding, we adopt a data-driven approach by leveraging a large dataset extracted from Wikipedia. The dataset serves as the foundation for generating a diverse set of negated sentences. Our approach incorporates dependency parsing and tense morphological structure to systematically construct negated counterparts of the original sentences. By explicitly introducing negation into the data, we aim to improve the model's ability to handle negated statements effectively.

The next step involves training the language models using a paired set of affirmative-affirmative and affirmative-negative sentence pairs. To effectively capture the nuances of negation, we employ a weighted loss function and an elastic weight regularizer during the training process. This combination enables the models to focus on learning from both affirmative and negative instances, thereby reducing any inherent model bias towards affirmative statements. To evaluate the effectiveness of our models, we conduct experiments on standard datasets, specifically focusing on knowledge completion and NLI tasks.

The work presented here addresses the shortcomings of negation understanding of LMs and makes the following contributions:

1. We provide a methodology to automatically

generate negated data from the original wiki corpus using dependency parsing, providing a diverse and representative source for negation understanding.

2. We propose a pre-training framework with Elastic Weight Consolidation (EWC) regularization and weighted loss to overcome catastrophic forgetting that aids to augment the negation knowledge in LMs.

3. We demonstrate that state-of-the-art transformers are not robust to negation and provide empirical evidence that negation augmented LMs outperform the classical LMs on original as well as negation datasets.

## 2   Related Work

Language models such as BERT (Devlin et al., 2019) and RoBERTa (Liu et al., 2019) have achieved remarkable results in various natural language understanding tasks such as question answering, sentiment analysis, named entity recognition (Gillard et al., 2006; Naik et al., 2018), etc. Their ability to capture contextual information, leverage large scale pre-training and the incorporation of novel training techniques have contributed to their impressive performance. These models also learn broader range of factual and common-sense knowledge (Akyurek et al., 2022). Despite these abilities and performance, Kassner and Schütze (2020) shows that these models fall short on understanding the negated factual sentences.

Various works investigate the linguistic knowledge captured in these LMs. The authors (Warstadt and Bowman, 2019) investigate the grammatical knowledge of the LMs and conclude that these models have near human performance on simple sentences and fail to make fine-grained grammatical distinctions. Authors (Marvin and Linzen, 2018) propose a dataset for evaluating the knowledge of understanding grammatically correct as well as incorrect sentences. The oLMpics work investigates BERT and RoBERTa model on the reasoning tasks which require comparison, conjunctions and composition. Other studies include diagnosing syntactic heuristics in NLI tasks (McCoy et al., 2019), inner workings of these models on (negative) polarity items (Jumelet and Hupkes, 2018) and discovering of NLP pipelines in BERT (Tenney et al., 2019).

Some recent work studies negation understanding in the area of negation detection (Khandelwal and Sawant, 2020), negation scope detections (Fan-

cellu et al., 2016; Morante and Daelemans, 2009; Li and Lu, 2018), attention analysis in negation scope (Zhao and Bethard, 2020) and focus detection (Shen et al., 2019). Authors (Naik et al., 2018) study linguistic phenomenons such as antonyms, negation, and spelling mismatch and find that these models rely on syntactic clues to make the NLE inferences. Hossain et al. (2020) proposes a benchmark for natural language inference tasks in a form of a text-hypothesis pair in which negation plays a critical role. Also, the authors find that the current state-of-the-art transformers struggle to understand negation which often results in wrong inferences. Noji and Takamura (2020) shows the utilities of explicit negative examples to improve the syntactic abilities of models for the NLI tasks.

The studies (Hossain et al., 2020; Laverghetta Jr. and Licato, 2022) show that these state-of-the-art transformer models struggle to understand the negation and there exists the need to inject negation understanding in these models. These works investigate the negation understanding of LMs. However, only a few works try to fix this negation bias in LMs. Hosseini et al. (2021) augments negation understanding in BERT by modelling objectives with an unlikelihood based on negated generic sentences. The proposed method successfully injects knowledge into BERT-base, however, fails in BERT-large model.

## 3   Dataset

In order to effectively augment negation, we need a dataset for affirmative and negative sentences compatible with LM's input requirements. However, a publicly available dataset with such characteristics is not available. An alternative approach is undertaken to extract the wiki-raw-text corpus and negate the sentences.

### 3.1   Wiki-raw-text corpus

Each page within the wiki-raw-text corpus (Merity et al., 2016) is analysed and the first two lines of text from each page is selected. The rationale behind the selection is that the first line typically provides an introductory overview or summary of the page's content, whereas the second line serves as a continuation or a follow-up statement. By utilizing this approach a collection of sentence pairs are obtained, where the first line represents an anchor sentence and the second line serves as the follow up sentence.

To prepare the dataset for training, pre-processing and cleaning steps are performed to ensure the dataset's quality and consistency. One specific step involves the removal of pronunciations and respelling keys. These keys are often found in text derived from the wikipedia corpus and provide phonetic indications for pronunciation variations. These keys are eliminated to ensure that the focus is solely on the textual content, without the inclusion of phonetic annotations or pronunciation indicators. Through this meticulous process, a comprehensive dataset of 25000 sentence pairs is generated. Next, we describe the method for negating an affirmative sentence.

## 3.2 Negation

Consider an affirmative sentence that needs to be converted into its negated form. We generate the negated form of a sentence using the dependency parsing tree and syntactical rules. The methodology for negating a sentence is as follows.

**Part of Speech tagging:** In order to analyse the sentence structure and facilitate further linguistic processing for tense extraction, the Part-of-Speech (POS) tags of the given sentence are obtained using a POS tagging algorithm[1]. POS tagging assigns specific grammatical labels to each word in a sentence which indicates its syntactic category and functions within the sentence. By labelling the words with their respective POS tags, the sentence is represented in the form of word/POS tag pairs, allowing for a structured representation of the sentence's linguistic patterns.

**Tense Patterns:** To accurately determine the tense of each sentence a systematic approach is employed involving the utilization of POS tags represented in the form of word/POS pairs. To achieve this, a set of predefined patterns for each tense type is specified. These patterns are designed to capture specific linguistic structures associated with different tenses. For instance, to identify *past* tense verbs, a pattern such as "\w+VBD" is defined, where "\w+" represents one or more word characters and "VBD" corresponds to the POS tag for past tense verbs. Similarly, other tense patterns are devised, such \w+VBP for *present* tense verbs and \w+MD for *modal* verbs. We provide in Appendix A.1 the full list of patterns for each tense. Subsequently, by utilizing regular expressions (regex), the predefined patterns are matched to the word/POS tag pairs of

each sentence, which allows for the extraction of the corresponding tense. For example, if the sentence contained the word/POS pair "walked/VBD", it would match the pattern for *past* tense sentences. This approach is generalized and facilitates the automated identification of the tense in sentences, enabling subsequent linguistic analysis and supporting the various other NLP tasks such as temporal information, extraction, text summarization, sentiment analysis and more.

**Conversion:** Following the detection of the tense in each sentence, a negation transformation process is implemented to convert the sentences into their negated forms. This transformation involves the addition, replacement and lemmatization of words within the sentence to convey a negated meaning. To introduce the negation cues, specific modifications are made to the sentence structure. For instance, in the case of past tense the main verb is first lemmatized and the negation cues are added before it. Consider an example "*She walked to the park.*" After negation transformation, it becomes "*She did not walk to the park.*" where "did not" serves as the negation cue inserted before the lemmatized main verb "walk". This effectively negates the original positive assertion. Additionally, we consider special keywords such as "somewhere", which are transformed into "nowhere" to express negation. We provide the list of special keywords in Appendix A.1. By applying these negation transformations systematically, the original sentences are modified to represent the negated statements, facilitating the analysis of negation and its impact on the semantic content and context of the text.

We illustrate below some of the examples in Table 1. In addition, to evaluate the negation, we manually verify 100 random negated sentences. The evaluation is made by English native speakers resulting in an overall precision of 96%. In general, the method can also be applied to other cases to generate negative samples.

## 3.3 Sample Generation

Based on the extracted two lines (first sentence, follow-up sentence) from each page of the wiki-raw-text corpus, two types of data are created. The first type retains the original format, where the sentence pairs consist of the affirmative sentence accompanied by the corresponding follow-up or continuous statement. The data serves as a control group for training the model with affirmation sen-

---

[1] https://spacy.io/usage/linguistic-features/

| Derived Tense | #Sentences | Example Affirmative Sentence | Converted Negative Sentence |
|---|---|---|---|
| Simple Past | 11186 | The episode premiered on the Fox network on October 7. | The episode did not premiere on the Fox network on October 7. |
| Simple Present | 1397 | Brockton is located approximately 25 miles of Boston. | Brockton is not located approximately 25 miles of Boston. |
| Modular verb | 493 | Searching a specific search tree according to a binary key can be recursively or iteratively programmed. | Searching a specific search tree according to a binary key cannot be recursively or iteratively programmed. |

Table 1: Summary of few derived tense category from the dataset of 25000 sentences, number of sentences along with examples of affirmative and their corresponding converted negated sentences.

tences. The second type of data involves negating the second sentence using the techniques described earlier. It allows for the examination of the effects of negation on the overall meaning and context of the sentence pairs. The first sentence remains unchanged as the affirmative statement, while the second sentence is now represented as the negated form of the original follow-up statement. The approach facilitates the creation of two distinct types of datasets: one representing the original unaltered sentence pairs and the other representing the original sentence with negated follow-up statements. We present the process in Figure 1.

## 4 Model

In this section, we describe the training sample generation and the model details.

### 4.1 Training Samples

The training data consists of the following samples:
1. $\langle S_a, S_f \rangle$ are unaltered sentence pairs where $S_a$ is the affirmative sentence and $S_f$ is the follow-up sentence.
2. $\langle S_a, S_{fn} \rangle$ where $S_a$ is the affirmative sentence and $S_{fn}$ is the negated sentence of $S_f$.

**Masking:** In each pair, we mask the tokens in the follow-up sentence (second sentence). The first sentence acts as an anchor sentence and gives context while predicting the MASK token in the second sentence.

The strategy for masking involves replacing either the object or subject of a sentence with special token "[MASK]", to indicate that the entity has been masked. The technique allows for a more focused analysis on the impact of negation on specific components of the sentence.

When masking the object of a sentence, the direct object is identified and replaced with the "[MASK]" token. For example, consider the original sentence "John ate an apple". In this case, the object "apple" is masked resulting in the modified sentence "John ate an [MASK].". This masking allows us to isolate the effect of negation on the object component of the sentence while keeping the rest of the sentence intact.

However, in situations where the object is not explicitly mentioned or unavailable, the subject of the sentence is masked instead. For instance, consider the sentence "Cat is sleeping". Since there is no explicit object in the sentence, the subject "Cat" is masked, leading to the transformed sentence "[MASK] is sleeping". The masking technique enables the examination of the effect of affirmation as well as negation on the subject component. Overall, masking helps to understand the semantic meaning and interpretation of a sentence. Note that, we do not mask any token of $S_a$. It acts as grounding (context) for the next sentence.

**Next Sentence Prediction:** In certain models, that incorporate the Next Sentence Prediction (NSP) task, specific labelling is required to indicate the relationship between pairs of sentences. To address this requirement, the following labelling scheme is implemented:

1. For pairs of affirmative sentences $\langle S_a, S_f \rangle$, a label 0 is assigned. It indicates that the two sentences are affirmative and are contextually related.

2. For pairs consisting of an affirmative and a negative sentence $\langle S_a, S_{fn} \rangle$, a label of 1 is assigned. This label signifies that the two sentences have a contrast in meaning or sentiment.

The labelling scheme enables the NSP task to effectively capture the relationship and context between pair of sentences. It helps in training and evaluating the model's ability to capture the semantic relationship and contextual nuances between sentences, particularly in scenarios involving negation. We explicitly imply the model for the following learning: (a) Given an affirmative sentence $S_a$ as context, predict the [MASK] token of the next affirmative sentence $S_f$ such that $S_f$ is the actual sentence that follows $S_a$. (b) Given an affirmative sentence $S_a$ as context, predict the [MASK] token of $S_{fn}$ (negated sentence of $S_f$) such that negated sentence is not the actual sentence that follows the sentence $S_a$.

### 4.2 Regularization

Fine-tuning language models is the most critical part of the transfer learning. The aggressive fine-

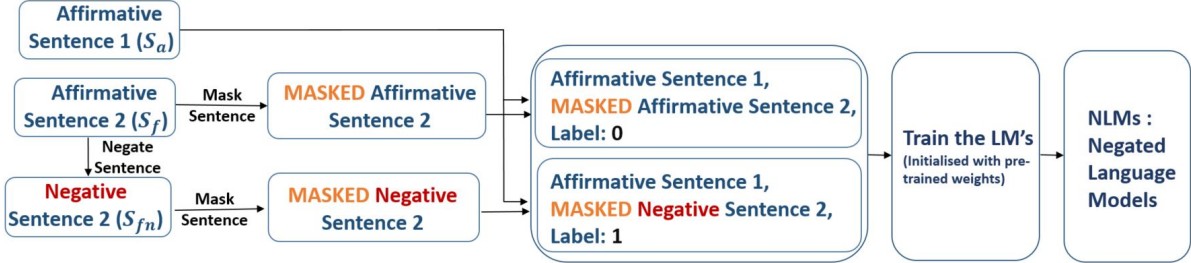

Figure 1: Data generation affirmative-affirmative, affirmative-negation pairs followed by training models.

tuning causes catastrophic forgetting, eliminating the knowledge of the information captured through previous learnings. On the other hand, too cautious fine-tuning leads to slow convergence and leads to overfitting. To eliminate these challenges, we deploy the strategies of freezing the layers and controlled change in the set of biases and weights $\theta$ to optimize performance and learning. Rather than fine tuning all the layers, we first propose to update only the top layers in models. Secondly, we apply EWC, a regularization technique to control forgetting specifically by constraining the model weights that are important for previous tasks. The sensitivity of the model with respect to its weight $\theta_i$ can be estimated by looking at the curvature of the loss surface along the direction of change in $\theta_i$. The high curvature signifies the slight $\theta_i$ change, resulting in sharp change in loss. To stop the weights of the model deviating from the original base model, we constrain the parameters using elastic weight consolidation (Kirkpatrick et al., 2017). Learning a task, consists to adjust the diagonal of Fisher Information matrix $F$, which signifies the second derivative of the loss near minimum. The $k^{th}$ diagonal element in $F$ denotes the importance of weight $\theta_i$. The intuition is to move the important weights (from the actual model) as little as possible, when the model is fine-tuned on the negation learning task. It is achieved by adding a regularization term to the loss function and is shown below.

$$L_{EWC} = L_\theta + \frac{\lambda}{2} \sum F_i(\theta_i - \theta_i')^2$$

Here $L_\theta$ denotes the loss, $\theta_i$ denotes the original model weights, $\theta_i'$ denotes the new model weights while training and $\lambda$ is the importance factor.

### 4.3 Loss

Next Sentence Prediction (NSP) and Masked Language Modelling (MLM) are the two most popular modelling tasks. To leverage the strengths of both the NSP and MLM tasks, we propose a weighted loss functions that combines the objective of both tasks. The idea is to assign appropriate weights to each task's loss component based on their respective importance and impact on model performance. It enables the model to capture both sentence-level representation and word-level representation leading to improved language understanding and generation capabilities.

Let $L_{NSP}$ be the loss associated with the NSP task and $L_{MLM}$ denotes the loss associated with the MLM task. We introduce the weighting factor $\alpha \in [0, 1]$ to balance the contributions of each loss component. We denote the total loss with $L_\theta$ and is defined below.

$$L_\theta = \alpha L_{MLM} + (1 - \alpha) L_{NSP}$$

We augment negation in BERT and RoBERTa language models with the above defined regularization and loss function. We describe the full training process in Figure 1.

## 5 Experiments

### 5.1 Evaluation Dataset

**Language Model Analysis (LAMA):** We use the LAMA probe (Petroni et al., 2019) in our experiments, which has emerged as a valuable dataset for evaluating factual knowledge. The evaluation dataset is a collection of cloze-style questions about real-world facts. The statements are facts or common-sense knowledge generated from either subject-relation-object triples (X, rel, Y) or question-answer pairs. LAMA leverages the mask set-up, where the model presented with a statement is required to predict the MASK token with a single token answer. The statements are generated by following a template for each relation, which include place-holders X and Y. Here, X is the subject and Y is replaced by the MASK token. For example, for triples ⟨iOS, developed, Apple⟩, the statement is iOS is developed by Apple. After masking, the statement iOS is developed by [MASK]

and is fed into the model to predict the MASK token with the ground truth value being `Apple`. LAMA is composed of various datasets. Google-RE[2] consists of three relations, namely "place-of-birth", "date-of-birth" and "place-of-death". T-REx (Elsahar et al., 2018) is a subset of wikipedia data triples which consists of 41 relations. ConceptNet consists of 16 relations from conceptNet knowledge base (Li et al., 2016). Squad is a subset of 305 context-insensitive questions manually rephrased as cloze style questions (Rajpurkar et al., 2016). We summarize the LAMA dataset in Appendix A.3.1.

**Negated LAMA:** Negated LAMA (Kassner and Schütze, 2020) was created by manually inserting the negation cues in the template of the LAMA dataset.

**Natural Language Inference Dataset:** We evaluate the pre-trained models on RTE (Dagan et al., 2005; Giampiccolo et al., 2007; Bentivogli et al., 2009) , SNLI (Bowman et al., 2015) and MNLI (Williams et al., 2018) datasets. As negation is under-represented (Table 16 in Appendix for reference) in these datasets, we also consider the negation dataset of these tasks from Hossain et al. (2020).

## 5.2 Experiment Setups

We aim to investigate the effects of augmenting negation in BERT and RoBERTa models of various sizes. Prior research (Hosseini et al., 2021), indicates that incorporating negation understanding into models with different sizes can present challenges and may yield varying results.

In our experiments, we employed state-of-the-art transformers BERT and RoBERTa models of varying sizes, namely BERT-base, BERT-large, RoBERTa-base and RoBERTa-large.

By including models of different sizes, we aim to study how the size of the underlying large language models affects the performance and adaptability of the negation augmentation technique. Note that the training approach requires careful calibrations as the objective is not only to augment the negation knowledge in these language models but also to preserve the previous learnings on affirmation sentences. We do the experiments with several settings and conclude to train the pre-trained models for 2 epochs with 5000 examples. We update top 5 layers of base models and top 10 layers of the

large models. We choose a maximum length of 128 and use AdamW as an optimizer with a linear rate scheduler. We apply EWC regularization, to conserve the previous learning as well as constrain the deviation from the original pre-trained weights. In addition, we use the weighted loss to train the models as explained in Section 4.3. We manually experiment with different sets of hyperparameters in the experiments and refer readers to Table 17 in Appendix for the detailed final list of hyperparameters. All the models are trained on two 24 GB P40 GPUs.

## 5.3 Results

Can the negated augmented transformers, outperform and solve the LAMA, negated-LAMA and the text-hypothesis pairs including negation as compared to the original models? Yes, they can.

**Knowledge Base Completion:** We evaluate the knowledge base completion task on LAMA and negated-LAMA datasets. In order to evaluate the proposed negation augmentation, we first evaluate the models on negated LAMA datasets. Following Hosseini et al. (2021), we report precision at k = 1 (higher is better) for original LAMA queries and mean top error rate for negated LAMA queries (lower is better). We report the results for the negation cases for BERT-base and BERT-large in Tables 3 and Table 5. NBERT and NRoBERTa denotes the negated augmented language models (NLMs). We make the following observations.

The augmented negation model for Bert-base outperforms the original BERT model by a margin of 8% and Hosseini et al. (2021) by a margin of 2%. For BERT-large models, the method outperforms the original BERT-large and the state-of-the-art model by 8%. As there exist no previous works on the augmentation of negation in RoBERTa models, we compare only the original models in Table 7 and Table 9. The proposed methodology outperforms the original Roberta-base model by 4% and the original Roberta-large model by 9%. Note that, the previous approaches BERTNOT (Hosseini et al., 2021) were not able to augment negation in larger models. However, the proposed technique effectively augments negation understanding in the base as well as large models. We also report the results for the affirmation cases in BERT-base, BERT-large, RoBERTa-base and RoBERTa-large in Tables 2, 4, 6 and 8. We find that augmenting negation understanding in these models has a subtle effect on affirmative LAMA queries and in most of the cases

---

[2]https://github.com/google-research-datasets/relation-extraction-corpus

| Model/LAMA | SQuAD | ConceptNet | TREx | GoogleRE |
|---|---|---|---|---|
| BERT-base | 13.53 | 15.65 | 29.10 | 10.24 |
| BERTNOT-base-KL[3] | 13.64 | 15.64 | 29.28 | 10.27 |
| BERTNOT-base [3] | 13.97 | 15.49 | 29.25 | 10.31 |
| NBERT-base | **11.14** | **14.27** | **30.02** | **10.44** |

Table 2: Mean precision at k=1(p@1) for original LAMA queries (higher is better) of proposed NBERT-base model with baselines.

| Model/NegLAMA | SQuAD | ConceptNet | TREx | GoogleRE |
|---|---|---|---|---|
| BERT-base | 8.61 | 2.24 | 21.42 | 3.76 |
| BERTNOT-KL[3] | 4.97 | 1.19 | 21.77 | 3.99 |
| BERTNOT [3] | 2.10 | 0.73 | 11.86 | 1.10 |
| NBERT-base | **1.31** | **0.66** | **2.39** | **0.06** |

Table 3: Comparision of Mean top 1 error rate for negated LAMA queries (lower is better) of proposed NBERT-base model with baselines.

| Model /LAMA | SQuAD | ConceptNet | TREx | GoogleRE |
|---|---|---|---|---|
| BERT-large | 16.83 | 19.26 | 30.76 | 10.93 |
| BERTNOT-large [3] | 14.19 | 19.14 | 32.09 | 11.02 |
| NBERT-large | **17.04** | **18.33** | **32.37** | **10.33** |

Table 4: Mean precision at k=1(p@1) for original LAMA queries (higher is better) of proposed NBERT-large model with baselines.

| Model /NegLAMA | SQuAD | ConceptNet | TREx | GoogleRE |
|---|---|---|---|---|
| BERT-large | 7.95 | 1.67 | 22.97 | 4.13 |
| BERTNOT-large [3] | 8.28 | 1.87 | 23.49 | 4.22 |
| NBERT-large | **0.98** | **1.62** | **0.53** | **0.01** |

Table 5: Comparison of Mean top 1 error rate for negated LAMA queries (lower is better) of proposed NBERT-large model with baselines.

| Model /LAMA | SQuAD | ConceptNet | TREx | GoogleRE |
|---|---|---|---|---|
| RoBERTa-base | 11.51 | 5.16 | 16.44 | 4.38 |
| NRoBERTa-base | **9.86** | **4.78** | **19.74** | **3.51** |

Table 6: Mean precision at k=1(p@1) for original LAMA queries (higher is better) of proposed NRoBERTa-base model with baselines.

| Model /NegLAMA | SQuAD | ConceptNet | TREx | GoogleRE |
|---|---|---|---|---|
| RoBERTa-base | 8.55 | 2.04 | 17.75 | 1.20 |
| NRoBERTa-base | **4.27** | **1.49** | **9.19** | **0.02** |

Table 7: Comparison of Mean top 1 error rate for negated LAMA queries (lower is better) of proposed NRoBERTa-base model with baselines.

| Model /LAMA | SQuAD | ConceptNet | TREx | GoogleRE |
|---|---|---|---|---|
| RoBERTa-large | 18.09 | 6.91 | 21.17 | 3.94 |
| NRoBERTa-large | **16.77** | **7.28** | **24.03** | **3.37** |

Table 8: Mean precision at k=1(p@1) for original LAMA queries (higher is better) of proposed NRoBERTa-large model with baselines.

| Model /NegLAMA | SQuAD | ConceptNet | TREx | GoogleRE |
|---|---|---|---|---|
| RoBERTa-large | 11.51 | 2.14 | 21.03 | 1.49 |
| NRoBERTa-base | **0.0** | **0.04** | **0.01** | **0.0** |

Table 9: Comparison of Mean top 1 error rate for negated LAMA queries (lower is better) of proposed NRoBERTa-large model with baselines.

| Sentence | Top-3 Prediction by BERT | Top-3 Prediction by NBERT |
|---|---|---|
| The capital of South Korea is [MASK]. | Seoul, Tokyo, Korea | Seoul, South, Ha |
| The capital of South Korea is not [MASK]. | **Seoul**, Known, listed | disputed, Olympics, merged |
| Charles Nodier died in [MASK] | Paris, Rome, Office | Paris, Rome, Office |
| Charles Nodier did not die in [MASK] | **Paris**, Rome, Office | Poverty, France, death |

Table 10: Example showcasing the prediction of BERT-base and NBERT-base. The wrong prediction by BERT is highlighted in bold.

| Test Pairs | RTE | | | SNLI | | | MNLI | | |
|---|---|---|---|---|---|---|---|---|---|
| | MB | BERT-base | NBERT-base | MB | BERT-base | NBERT-base | MB | BERT-base | NBERT-base |
| Original | | | | | | | | | |
| dev | 52.7 | 66.1 | 71.14 | 33.8 | 89.9 | 90.65 | 35.5 | 83.2 | 83.4 |
| dev neg | 51.2 | 63.4 | 56.1 | 54.4 | 89.4 | 90.78 | 50.2 | 83 | 82.8 |
| New w/ neg. | | | | | | | | | |
| Tneg-H | 80.2 | 65.2 | 72.8 | 62 | 32.6 | 38 | 45.8 | 65.6 | 64.2 |
| T-Hneg | 91 | 39.2 | 84.6 | 41 | 58.8 | 60.6 | 47.6 | 62.4 | 65.6 |
| Tneg-Hneg | 65.6 | 68.4 | 56 | 69.8 | 41.8 | 35.8 | 47 | 63.6 | 61.4 |
| All | 78.9 | 57.6 | 71.14 | 57.6 | 44.4 | 44.8 | 46.8 | 63.9 | 63.8 |

Table 11: Comparison of state-of-the-art models trained with the original training split for each benchmark and evaluated with - the original development split (dev), pairs in the original development split containing negation (dev neg), and the new pairs containing negation (New w/neg). MB stands for the majority baseline, BERT-base is the original base model and NBERT-base model is the proposed negated augmented BERT-base model.

| Model /LAMA | SQuAD | ConceptNet | TREx | GoogleRE |
|---|---|---|---|---|
| **NBERT-base** | 11.14 | 14.27 | 30.02 | 10.44 |
| Without EWC | 11.47 | 14.00 | 28.54 | 9.42 |
| L2 Regularization | 11.47 | 13.93 | 26.93 | 9.01 |
| MLM | 10.16 | 13.99 | 28.12 | 8.49 |
| MLM + NSP | 12.13 | 13.78 | 28.55 | 9.43 |
| Train data-size - 10000 | 12.78 | 14.79 | 27.97 | 7.07 |
| Train datasize - 20000 | 10.16 | 14.77 | 28.95 | 7.38 |
| All layers updates | 11.47 | 13.93 | 26.93 | 9.01 |

| Model /negLAMA | SQuAD | ConceptNet | TREx | GoogleRE |
|---|---|---|---|---|
| **NBERT-base** | 1.31 | 0.66 | 2.39 | 0.06 |
| Without EWC | 0.98 | 0.52 | 1.96 | 0.09 |
| L2 Regularization | 1.31 | 0.66 | 1.83 | 0.06 |
| MLM | 0.98 | 0.42 | 1.30 | 0.03 |
| MLM + NSP | 1.97 | 0.79 | 2.78 | 0.09 |
| Train data-size - 10000 | 3.94 | 1.35 | 12.37 | 0.20 |
| Train datasize - 20000 | 4.60 | 1.68 | 14.95 | 0.17 |
| All layers updates | 1.31 | 0.66 | 1.83 | 0.06 |

Table 12: Ablation study signifying the impact of the regularizers namely EWC and L2, different loss functions, data size and all layers update. The left table shows the results on original LAMA queries (Higher is better) and the right table shows the result on negated LAMA queries (Lower is better).

performs at par with the original models. We observe that RoBERTa-large model is more adaptable to negation understanding as compared to other models. We also showcase a few examples of the comparative results between BERT and NBERT for the knowledge base completion task. As showcased in Table 10, the BERT model is not able to understand negation and predicts wrong completion whereas the NBERT predicts the tokens correctly.

**Natural Language Inference:** We also evaluate the proposed models on RTE, MNLI and SNLI tasks. We fine-tune the models on original development splits and the new splits from Hossain et al. (2020) which consists of negation for each task. We use the same set of hyper-parameters from Hossain et al. (2020) to fine-tune models for BERT-base and RoBERTa-base. We summarize the results for BERT-base models in comparison to state-of-the-art models in Table 11. The fine-tuned NBERT-base model achieves superior results on original dev splits on RTE and SNLI tasks as compared to BERT-base and majority baseline. For negative dev splits, we observe the NBERT-base model outperforms the original models on SNLI tasks and performs poorly on RTE and on-par on MNLI datasets. For the new negation pairs, the NBERT-base model outperforms the original BERT-base-model on RTE and SNLI tasks and almost at par on MNLI tasks. We also evaluate BERT-large, RoBERTa-base and RoBERTa-large models and illustrate the results in Appendix A.4.2. We find similar observations for Roberta models. The experiment shows that the augmentation of negation understanding outperforms the original models in most of the downstream tasks.

### 5.4 Ablation Study

To test the effectiveness of our approach, we ablate the individual cases with a fixed set of other parameters. As shown in Table 12, our training approach setting is effective as compared to other variations. We make the following observation. In individual cases such as without using EWC regularization, L2 regularization, only MLM, equal weight for MLM and NSP (same as original BERT) and updating all the layers, we find that the accuracy drops for LAMA queries. Whereas, using greater data sizes namely 10000 and 15000 leads to an increase in error for negated LAMA queries. It is intuitive, as a larger number of negation samples may bias the models towards negation and lead to forgetting of affirmation understanding. The study signifies the importance of optimization to minimize the deviation from the original model as well as augment negation understanding in the model.

## 6 Conclusion

Negation is ubiquitous in a language and plays a crucial aspect in language understanding. We introduce a negation aware technique that addresses the negation bias of the existing LMs. We modified the training data by augmenting it with explicitly negation data and trained the model with EWC regularization with weighted loss. By incorporating, explicit negation cues during training, the proposed approach enables the model to better capture the subtle nuances of negation, thus resulting in improved contextual understanding and more accurate language completion. The results demonstrate that NLMs significantly improve the LMs performance in understanding and representing negated contexts, leading to more accurate predictions in knowledge completion and NLI tasks.

In the future, we plan to augment negation in cross-lingual and generative models. Augmenting negation in these models involves several challenges such as different syntactic and lexical cues, inherent difficulty of aligning and training paradigms.

## Limitations

While our approach yields promising results in improving negation understanding and mitigating modelling bias, we acknowledge a limitation in the case of tiny models. The tiny models are low-capacity models with limited parameters, designed to be lightweight and efficient in terms of computational nuanced understanding of a language model. Due to the constrained architecture, these models struggle to effectively capture and generalize the negation augmented tiny model on LAMA and negated LAMA dataset summarized in Table 23 and Table 24. We find that augmenting negation not yields substantial improvements in their performance.

Also, we limit our work to encoder-based models. However, there requires further exploration to understand the embedding of negation knowledge in GPT-based generative models. By addressing these limitations, future work pave the way for a more comprehensive and effective understanding of negation in language models.

## Ethics Statement

The research work complies with the EMNLP ethics policy and does not raise any ethical concerns.

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

# A  Appendix

We provide the details on tense patterns and the corresponding actions to convert affirmative to negative sentences in Section A.1. Section A.2 gives the details of hyperparameters to train the model and to fine-tune the model on NLI tasks. We give more details on the evaluation dataset for knowledge completion and NLI tasks in Section A.3. Section A.4 provides more examples from knowledge base completion tasks and benchmarks on NLI tasks for BERT-large, RoBERTa-base and RoBERTa-large.

## A.1  Tense Patterns

We summarize the pre-defined tense patterns for each tense in Table 14. We also give operations for each tense type to convert an affirmative to a negative sentence along with some examples. In Table 13, we provide the list of special words and their corresponding negated form. We replace the special words with the negated form to convert a sentence to its negated form.

| Tokens | Negated Form | Tokens | Negated Form |
|--------|--------------|--------|--------------|
| Ever | Never | Anybody | Nobody |
| Anyone | Noone | Anything | Nothing |
| Anywhere | Nowhere | Someone | Noone |
| Somebody | Nobody | Someone | None |
| Always | Never | Either | Neither |

Table 13: List of special tokens and their negated form. The tokens are replaced by the negated form to convert the affirmative sentence to negative sentence.

## A.2  Reproducibility

### A.2.1  Hyper-parameters for augmenting negation

Table 17 lists all the hyper-parameters used in our fine-tunings approach for different models and size.

### A.2.2  Hyper-parameters for NLI tasks evaluation

We list the hyperparameters to fine tune the models on NLI tasks in Table 18.

## A.3  Evaluation Dataset

### A.3.1  LAMA dataset

We summarize the LAMA dataset in Table 15. LAMA consists of datasets from various sources namely Google-RE, TREx, ConceptNet and Squad.

### A.3.2  Natural Language Inference dataset

We provide below summary of percentage of negation in NLI benchmarks (Hossain et al., 2020).

## A.4  Additional Results

We provide some more examples of knowledge-base completion tasks for the BERT-large model. We also summarize the evaluation for NLI tasks on BERT-large, RoBERTa-base and RoBERTa-large models. Note that, for the RoBERTa-base model, we compare our results with state-of-the-art models. In the case of BERT-large and RoBERTa-large, we compare the negated augmented models with the original models.

### A.4.1  More Examples - Knowledge Base Completion tasks

We provide below some more examples for the BERT-large model in Table 19. Observe that the original BERT-large model predicts the same token for affirmative and negative sentences. The wrong predictions are highlighted in bold. The negated augmented NBERT-large model predicts the correct tokens.

### A.4.2  Benchmarks models on NLI tasks

We also benchmark the negation augmented trained models namely RoBERTa-base, NBERT-base and RoBERTa-large on RTE, SNLI and MNLI tasks on original splits as well as negation splits. We find that the proposed NBERT-large model outperforms the original BERT-large in all three tasks on original and negation splits and is shown in Table 20. For RoBERTa-base and large model, the negated augmented model performs as par as the original models and is shown in Table 21 and Table 22.

### A.4.3  Benchmark of BERT-tiny on LAMA and neg-LAMA datasets.

We also tried out the methodology to augment negation in BERT-tiny. We find that augmenting negation in the tiny model is challenging and the models augmented with negation perform poorly on LAMA as well as negated-LAMA queries. We summarize the results in Table 23 and Table 24 below. The work requires further investigations.

| Sentence Type | Pattern | Operation | Pattern Match | Example |
|---|---|---|---|---|
| Modular Verb | \w+/MD, \w+/MD + \w+/PRP | Add negation cues "not" after modular verb | wou;d/MD | In the late 1950s he would also serve briefly on the board of directors of another major , MGM. |
| Non-3rd person Present Tense | \w+/VBP | Add negation cue "do not" before verb. | use/VBP | They use cameras , camcorders and audio recorders to conduct overt surveillance of the public. |
| 3rd person Present Tense | \w+/VBZ | Add negation cue "does not" after verb. | links/VBZ | The route links Interstate 10 midway between the Coachella Valley and Blythe on the California |
| Past Tense | \w+/VBD | Add negation cue "not" after verb. | was/VBD | Built in 1840 , it was part of Little Rock 's first military installation |
| Past Participle | \w+/VBN | Add negation cue "did not" before lemmatized verb. | qualified/VBN | Through their performance in the Twenty20 Cup , the qualified for the Champions League Twenty20. |
| Verb base form | \w+/VB | Add negation cue "not" before verb. | use/VB | Several companies , mostly retailers , use the company 's services to showcase products |
| Proper Singular, Plural | \w+/NN[P]? + /w+/NNS | Add negation cue "does not" after verb. | production/NN stars/NNS | The Rainforest Films production stars Idris Elba , Beyoncé , and Ali Larter |

Table 14: Examples of pre-defined tense patterns for various sentence type. The operation signifies the different actions based on sentence type to convert a affirmative to negative sentence. The Pattern Match column indicates the matched pattern from the example column.

| Corpus | Relation | Statistics | |
|---|---|---|---|
| | | Facts | Relations |
| Google-RE | birth-place | 2937 | 1 |
| | birth-date | 1825 | 1 |
| | death-place | 765 | 1 |
| | Total | 5527 | 3 |
| T-REx | 1-1 | 937 | 2 |
| | N-1 | 20006 | 23 |
| | N-M | 13096 | 16 |
| | Total | 34039 | 41 |
| ConceptNet | Total | 2996 | 16 |
| Squad | Total | 305 | - |

Table 15: Statistics of LAMA dataset.

| NLI Benchmarks | #Sentences | % w/ negation |
|---|---|---|
| RTE | 16389 | 7.16 |
| SNLI | 1138598 | 1.19 |
| MNLI | 883436 | 22.63 |

Table 16: The proportion of sentences containing negation is lower than English language corpora(approximately 25%) in existing natural language inference benchmarks. This is particularly true for the RTE and SNLI datasets.

| Parameters | BERT-base-cased | BERT-large-cased | RoBERTa-base-cased | RoBERTa-large-cased | Bert-tiny |
|---|---|---|---|---|---|
| Batch Size | 64 | 64 | 64 | 64 | 64 |
| Epoch | 2 | 2 | 2 | 2 | 2 |
| Max Tokens | 256 | 256 | 256 | 256 | 256 |
| #Training Samples | 5000 | 5000 | 5000 | 5000 | 5000 |
| #EWC Samples | 30000 | 30000 | 30000 | 30000 | 30000 |
| EWC Importance | 1e-16 | 1e-16 | 1e-16 | 1e-16 | 1e-16 |
| # Layer Freeze | 5 | 10 | 5 | 10 | 1 |
| Learning Rate | 1e-4 | 5e-5 | 3e-4 | 1e-5 | 5e-5 |
| Adam Epsilon | 1e-8 | 1e-8 | 1e-8 | 1e-8 | 1e-8 |
| Clip Grad Norm | 1.0 | 1.0 | 1.0 | 1.0 | 1.0 |
| $\alpha$ | 0.8 | 0.8 | - | - | 0.8 |

Table 17: Hyperparameters for fine-tuning LMs to augment negation understanding.

| Hyperparameters | RTE | | SNLI | | MNLI | |
|---|---|---|---|---|---|---|
| | BERT-base/ BERT–large | RoBERTa-base/ RoBERTa-large | BERT-base/ BERT-large | RoBERTa-base/ RoBERTa-large | BERT-base/ BERT-large | RoBERTa-base/ RoBERTa-large |
| Batch size | 8 | 16 | 32 | 32 | 32 | 32 |
| Learning Rate | 2e-5 | 2e-5 | 1e-5 | 1e-5 | 2e-5 | 2e-5 |
| Epochs | 50 | 10 | 3 | 3 | 3 | 3 |
| Weight decay | 0.0 | 0.0 | 0.1 | 0.1 | 0.0 | 0.0 |

Table 18: Hyperparameters for fine-tuning models on RTE, SNLI, and MNLI tasks.

| Sentence | Top-3 prediction by BERT | Top-3 prediction by NBERT |
|---|---|---|
| A teacher is most likely teaching at a [MASK]. | school, university, college | school, university, college |
| A teacher is not most likely teaching at a [MASK]. | **school, university, college** | age, school, funeral |
| Marcel Oopa died in the city of [MASK]. | Paris, Warsaw, Helsinki | Paris, Liège, Ljubljana |
| Marcel Oopa did not die in the city of [MASK]. | **Paris**, Warsaw, Helsinki | age, friendly, the |
| Bible is a [MASK] text. | religious, sacred, complete | religious, sacred, Christian |
| Bible is not a [MASK] text. | **religious, sacred, complete** | new, friendly, novel |

Table 19: The prediction of BERT-large and NBERT-large, wrong predictions are highlighted in bold.

| Test pairs | RTE | | SNLI | | MNLI | |
|---|---|---|---|---|---|---|
| | BERT-large | NBERT-large | BERT-large | NBERT-large | BERT-large | NBERT-large |
| Original | | | | | | |
| dev | 58.27 | 72.26 | 52.4 | 91.69 | 66.87 | 65.4 |
| dev neg | 46.34 | 51.21 | 60.2 | 91.24 | 55 | 70 |
| New w/ neg. | | | | | | |
| Tneg-H | 65.6 | 78.4 | 48.6 | 47.6 | 64.8 | 65.4 |
| T-Hneg | 40.4 | 80.4 | 62.8 | 60.6 | 70 | 70 |
| Tneg-Hneg | 68.8 | 64 | 45.8 | 42.8 | 65.8 | 64.8 |
| All | 55.2 | 74.2 | 52.4 | 50.3 | 66.9 | 66.7 |

Table 20: Comparison of BERT-large models trained with the original training split for each benchmark and evaluated with - the original development split (dev), pairs in the original development split containing negation (dev neg), and the new pairs containing negation (New w/neg).

| Test Pairs | RTE | | | SNLI | | | MNLI | | |
|---|---|---|---|---|---|---|---|---|---|
| | MB | RoBERTa-base | NRoBERTa-base | MB | RoBERTa-base | NRoBERTa-base | MB | RoBERTa-base | NRoBERTa-base |
| Original | | | | | | | | | |
| dev | 52.7 | 75.8 | 73.58 | 33.8 | 91.6 | 91 | 35.5 | 87.9 | 88 |
| dev neg | 51.2 | 78.1 | 79.1 | 54.4 | 91.7 | 91.3 | 50.2 | 88 | 87.2 |
| New w/ neg. | | | | | | | | | |
| Tneg-H | 80.2 | 70.8 | 80 | 62 | 46.4 | 42.8 | 45.8 | 66.2 | 65.4 |
| T-Hneg | 91 | 51.4 | 91 | 41 | 63.6 | 65.4 | 47.6 | 70.4 | 69.2 |
| Tneg-Hneg | 65.6 | 65.4 | 66.2 | 69.8 | 45.8 | 44.2 | 47 | 63.6 | 64.2 |
| All | 78.9 | 62.5 | 79.0 | 57.6 | 51.9 | 50.8 | 46.8 | 66.7 | 66.27 |

Table 21: Comparison of RoBERTa-base models trained with the original training split for each benchmark and evaluated with - the original development split (dev), pairs in the original development split containing negation (dev neg), and the new pairs containing negation (New w/neg).

| Test Pairs | RTE | | SNLI | | MNLI | |
|---|---|---|---|---|---|---|
| | RoBERTa-large | NRoBERTa-large | RoBERTa-large | NRoBERTa-large | RoBERTa-large | NRoBERTa-large |
| Original | | | | | | |
| dev | 85.8 | 88.4 | 63.8 | 91.5 | 55.5 | 89 |
| dev neg | 73.17 | 80.49 | 64.4 | 92.3 | 60.2 | 89.2 |
| New w/ neg. | | | | | | |
| Tneg-H | 88 | 93.4 | 45.4 | 50.4 | 67.8 | 67.4 |
| T-Hneg | 87.6 | 90.6 | 66 | 70.4 | 73.6 | 74.2 |
| Tneg-Hneg | 81.8 | 81.2 | 44.2 | 48.8 | 67 | 67.2 |
| All | 85.8 | 88.4 | 51.9 | 56.5 | 69.5 | 69.6 |

Table 22: Comparison of RoBERTa-large models trained with the original training split for each benchmark and evaluated with - the original development split (dev), pairs in the original development split containing negation (dev neg), and the new pairs containing negation (New w/neg).

| Model /LAMA | SQuAD | ConceptNet | TREx | GoogleRE |
|---|---|---|---|---|
| BERT-tiny | 1.96 | 2.58 | 9.84 | 3.91 |
| NBERT-tiny | 1.38 | 2.04 | 1.25 | 0.0 |

Table 23: Mean precision at k=1(p@1) for original LAMA queries (higher is better) of proposed BERT-tiny model with baselines.

| Model /NegLAMA | SQuAD | ConceptNet | TREx | GoogleRE |
|---|---|---|---|---|
| BERT-tiny | 1.97 | 0.93 | 8.58 | 2.00 |
| NBERT-tiny | 1.38 | 0.30 | 0.94 | 0.0 |

Table 24: Comparison of Mean top 1 error rate for negated LAMA queries (lower is better) of proposed BERT-tiny model with baselines.