# OpenReview forum: "NLMs: Augmenting Negation in Language Models"
_EMNLP/2023/Conference — EMNLP 2023 Findings_

### Official Review · Reviewer_nWX2 · 2023-08-03

**Soundness:** 3

**Excitement:**

3: Ambivalent: It has merits (e.g., it reports state-of-the-art results, the idea is nice), but there are key weaknesses (e.g., it describes incremental work), and it can significantly benefit from another round of revision. However, I won't object to accepting it if my co-reviewers champion it.

**Paper Topic And Main Contributions:**

They investigate ways to continue the pre-training of pre-trained BERT models so that the resulting models are better able to handle negation. Their technique continues the pre-training for two epochs on 5000 examples that feature negative statements, avoiding "catastrophic forgetting" by incorporating a regularization term in the loss function and by restricting parameter updates to the top 5 layers of BERT-base and the top 10 layers of BERT-large models.

Their training examples consists of the first two sentences from Wikipedia articles, where sometimes the second sentence has been negated using a custom algorithm that employs part-of-speech tags and a dependency parse. Their pre-training loss function incorporates an MLM objective and a next sentence objective (just like the original BERT loss). However, their MLM objective only ever masks the object of the second sentence (or the subject if no object exists). The model is rewarded for correctly filling in the mask for the unaltered sentence pair, and is penalized for correctly filling in the mask for the negated sentence pair.

**Reasons To Accept:**

The results seem convincing.

**Reasons To Reject:**

The approach seems to be balanced on a knife edge, with very specific training techniques being necessary to make it work.

**Reproducibility:**

3: Could reproduce the results with some difficulty. The settings of parameters are underspecified or subjectively determined; the training/evaluation data are not widely available.

**Reviewer Confidence:**

2: Willing to defend my evaluation, but it is fairly likely that I missed some details, didn't understand some central points, or can't be sure about the novelty of the work.

**Typos Grammar Style And Presentation Improvements:**

sec 1: "do not appropriately models" => "do not appropriately model"
sec 1: "LMs when finetuned on..." => "When finetuned on..."
sec 2: "Language models such as BERT, RoBERTa" => "Language models such as BERT and RoBERTa"
sec 2: "leverage on large scale pre-training" => "leverage large scale pre-training"
sec 2: don't preface references with "The authors" or "Authors" or "The studies"
sec 2: "grammatical correct" => "grammatically correct"
sec 2: "on the reasoning tasks which requires" => "on reasoning tasks which require"
sec 2: "The other studies include" => "Other studies include"
sec 2: "Some recent work studies the negation understanding" => "Some recent work studies negation understanding"
sec 2: "to improve the syntactic abilities of the models" => "to improve the syntactic abilities of models"
sec 3: "public available" => "publicly available"
sec 3: "to extract the wiki-raw-text corpus and negate the sentences" => "to negate the sentences of the wiki-raw-text corpus"
sec 3.2: "consider an affirmative sentence, that needs" => "consider an affirmative sentence that needs"
sec 3.2: "the dependency parsing tree and the syntactical rules" => "the dependency parse tree and syntactic rules"
sec 3.2: ' "\w+VBD" is defined, where "\w" represents...' => ' "\w+/VBD" is defined, where "\w+" represents...'
sec 4.1: "between pair of sentences" => "between pairs of sentences"
sec 4.1: "we explicitly imply the model" => ???
sec 4.2: "The aggressive fine-tuning" => "Aggressive fine-tuning"
sec 4.2: When EWC is first used, expand the acronym.
sec 4.2: "Learning a task, consists of adjust the diagonal of Fisher Information matrix F, which signifies the second derivative of the loss near minimum" => ??? (hard to definitively know what this sentence is trying to say, but it needs to be re-expressed)
sec 4.3: "Mask Language Modelling" => "Masked Language Modelling"
sec 4.3: "To leverage, the strengths" => "To leverage the strengths"
sec 4.3: "we propose a weighted loss functions that combine" => "we propose a weighted loss function that combines"
sec 4.3: "We introduce the weighted factor \alpha" => "We introduce weighting factor \alpha"
sec 4.3: "with the above defined regularization and the loss function" => "with the above defined regularization and loss function"
sec 5.1: "which has emerged as a valuable data" => "which has emerged as a valuable dataset"
sec 5.1: "wikiedia" => "wikipedia"
sec 5.2: "Note that, the training approach" => "Note that the training approach"
sec 5.2: "to train the pre-trained models with 2 epochs" => "to train the pre-trained models for 2 epochs"
sec 5.2: "refer to readers" => "refer readers to"
sec 5.3: "for the affirmation cases" => "for the affirmative cases"
sec 5.3: "has a subtle effect" => "has a minor effect"
ethics: "The research work comply with EMNLP ethics policy and do not have any ethical concerns." => "The research work complies with the EMNLP ethics policy and does not raise any ethical concerns."

---

> ### Author Rebuttal · Authors · 2023-08-29
>
> We thank the reviewer for the comments and suggestions.
>
> 1. We agree that our approach does require a careful balance of training techniques to achieve the desired results. We believe that this precision is what allows our method to excel in mitigating forgetting in language models. By carefully fine-tuning the model's weights and utilizing strategies such as EWC, we are able to strike a balance between preserving the knowledge gained from the pre-training task and adapting to the negation understanding at hand. While it is true that this precision can be challenging to achieve, we believe that the benefits of our approach far outweigh the difficulties. We believe that the method can also be applied to generative models such as GPT and LLaMA with some optimization in parameters.
>
> 2. We will carefully address and correct all the grammatical errors suggested.
>
> 3. We provide a comprehensive list of all parameters used in both our training and evaluation experiments in appendix. To enhance the reproducibility of our work, we will provide a more detailed explanation of negation generation, computing infrastructure, and other relevant details.

---

### Official Review · Reviewer_f5qH · 2023-08-05

**Soundness:** 3

**Excitement:**

3: Ambivalent: It has merits (e.g., it reports state-of-the-art results, the idea is nice), but there are key weaknesses (e.g., it describes incremental work), and it can significantly benefit from another round of revision. However, I won't object to accepting it if my co-reviewers champion it.

**Paper Topic And Main Contributions:**

Although negation plays an important role across a wide range of applications,
they are under-represented in pretrained language models (LMs).
This paper proposes a language model objective with a weighted cross-entropy loss and elastic weight consolidation regularization for the augment negation understanding. For negated augmented models, they reduce the mean top 1 error rate for BERT base to l.1%, BERT-large to 0.78%, RoBERTa base to 3.74%, RoBERTa-large to 0.01% on the negated LAMA dataset that outperform the existing negation models. It minimizes the mean error rate by a margin of 8% and 6% for original BERT and RoBERTa models. They also provide empirical evidences that negated augmented models outperforms the classical models on original as well as negation benchmarks on natural language inference tasks.

**Questions For The Authors:**

In line 379, "Rather than fine tuning all the layers, we first propose to update only the top layers in models. Secondly, we apply EWC,
a regularization technique to control forgetting specifically by constraining the model weights that are important for previous tasks."
What is a rational behind this? If some comparison of this method and fine tuning all layers, I would like to know.

**Reasons To Accept:**

The proposal of their pre-training framework with Elastic Weight Consolidation (EWC) regularization and weighted loss is interesting. The data generation for this pre-training framework (Figure 1) is interesting, too.

Table 12 shows various comparison with different loss functions, which shows interesting experiments.

**Reasons To Reject:**

Table 11 shows results where RTE is good while SNLI and MNLI are so so.

Data generation process in Figure 1 are quite limited since they extracted two lines (first sentence, follow-up sentence) from each page of the wiki-raw-text corpus. Thus this generates tiny corpus. All the more, if this generation yields errors, the results are instantly affected.




**Reproducibility:**

4: Could mostly reproduce the results, but there may be some variation because of sample variance or minor variations in their interpretation of the protocol or method.

**Reviewer Confidence:**

3: Pretty sure, but there's a chance I missed something. Although I have a good feel for this area in general, I did not carefully check the paper's details, e.g., the math, experimental design, or novelty.

---

> ### Author Rebuttal · Authors · 2023-08-29
>
> We thank the reviewer for the comments and suggestions.
>
> 1. The NBERT model, after fine-tuning, demonstrates superior performance on downstream tasks, compared to the majority baseline. RTE consists of shorter sentences, which generally favor model's performance due to its greater ability to capture short-range dependencies as compared to long sentences. However, MNLI task features longer sentences, the model achieves nearly identical accuracy to baselines. This highlights the classical challenge of understanding larger sentences (affirmative as well as negative) in language models, which warrants further investigation.
>
> 2. We believe that data generation process is efficient and can generate huge pile of dataset. In our experiments, we intentionally opted for a smaller dataset to mitigate the risk of introducing bias in the understanding of negation and affirmation, which could have led to more drastic weight changes. We refer the reviewers to Table 12 where we present the results for 10000 and 20000 samples. The greater train data size leads to an increase in error in negated LAMA queries. Nevertheless, we also manually examine the negated generated sentences (96% precision (refer section 3.2)) and the training data to marginalize any error.
>
> 3. We refer to Table 12, that showcases the comparative performance of fine-tuning all layers versus employing other regularization techniques, specifically L2 regularization and without using Elastic Weight Consolidation (EWC). The results indicate that updating all layers leads to a decline in accuracy for LAMA queries. In contrast to other regularization technique, EWC effectively mitigates forgetting by constraining the model weights that are crucial in classical language models.
>
> 4. We provide a comprehensive list of all parameters used in both our training and evaluation experiments in appendix. To enhance the reproducibility of our work, we will provide a more detailed explanation of negation generation, computing infrastructure, and other relevant details.

---

### Official Review · Reviewer_ZAkP · 2023-08-06

**Typos Grammar Style And Presentation Improvements:** No
**Soundness:** 3

**Excitement:**

4: Strong: This paper deepens the understanding of some phenomenon or lowers the barriers to an existing research direction.

**Missing References:**

No

**Paper Topic And Main Contributions:**

This work focuses on addressing the issue of negation understanding in pretrained language models (PLMs). Negation is a crucial element in natural language that reverses the semantic meaning of a sentence. However, it is often inadequately represented in PLMs, leading to incorrect inferences in various applications.

To improve the understanding of negation, the authors propose a novel approach. They introduce a language model objective that includes a weighted cross-entropy loss and elastic weight consolidation regularization. This approach is designed to augment the LMs' capability to handle negation effectively.

The experiments conducted on the negated LAMA dataset show promising results. The proposed negated augmented models achieve significant error rate reductions compared to existing negation models. Specifically, for BERT-base, BERT-large, RoBERTa-base, and RoBERTa-large, the mean top 1 error rates are reduced to 1.1%, 0.78%, 3.74%, and 0.01%, respectively. These models outperform the original BERT and RoBERTa models by 8% and 6% in terms of minimizing the mean error rate.

Furthermore, the authors demonstrate that the negated augmented models also outperform classical models on both original and negation benchmarks (Tab.2 to Tab.12) for natural language inference tasks.

In summary, this work presents an effective approach to enhance the understanding of negation in pretrained language models, resulting in improved performance on negation-related tasks and natural language inference benchmarks.

**Questions For The Authors:**

1. Although the proposed method can mitigate the issue of not recognizing negation in LLMs, it is still hard to explain why the LLMs understand the negation. So do you have any solution for LLMs to explain the negation understanding?
2. The negation appears not only in LLMs but also Vision-Language Models. Can your proposed method be applied to these?

**Reasons To Accept:**

* The paper proposed a new weight consolidation regularization that can help the negation understanding better.
* The paper tested on multiple benchmarks demonstrating the consistency and power of the proposed method.

**Reasons To Reject:**

* The method is not explainable although it helps negation understanding.

**Reproducibility:**

3: Could reproduce the results with some difficulty. The settings of parameters are underspecified or subjectively determined; the training/evaluation data are not widely available.

**Reviewer Confidence:**

3: Pretty sure, but there's a chance I missed something. Although I have a good feel for this area in general, I did not carefully check the paper's details, e.g., the math, experimental design, or novelty.

---

> ### Author Rebuttal · Authors · 2023-08-29
>
> We thank the reviewer for the comments and suggestions.
>
> 1. We agree that the paper do not investigates the underlying reason that why language models understand negation. The study's primary objective is to enhance negation understanding in language models. Negation is underrepresented in training data. Intuitively, it is the reason that transformer models struggle to understand negation. The study of explainability in negated language models is a novel and interesting area of research that will help to understand interpretability aspects of these NLMs. We consider this aspect to be a future research direction.
>
> 2. Negation is omnipresent in both language and vision tasks. For instance, in the context of visual question answering, a language model must be able to comprehend negation when generating responses, especially when the question requires an understanding of negation in relation to the visual scene. Without negation understanding, the language model will fail to give the right responses. The study aims to enhance negation comprehension in language models. However, we propose that utilizing negated language models in cases of negated visual question answering tasks will yield better outcomes. This problem presents an interesting research opportunity, opening up new avenues for exploration.
>
> 3. We provide a comprehensive list of all parameters used in both our training and evaluation experiments in appendix. To enhance the reproducibility of our work, we will provide a more detailed explanation of negation generation, computing infrastructure, and other relevant details.

---

### Meta-Review · Area_Chair_tTnf · 2023-09-13

**Recommendation:** 4

**Metareview:**

The paper investigates negation understanding in LMs. It proposes fine-tuning models on data that includes negation understanding and propose a two-part loss to balance fine-tuning with avoiding catastrophic forgetting. Experiments show that this recipe improves negation understanding for MLMs without hurting performance on other tasks.

The reviewers agree that the proposed recipe is interesting and experiments support the hypotheses well. There are some concerns regarding generalization to longer sequences and whether the proposed approach would generalize beyond the experiments in this work, both of which the authors acknowledge would be interesting future work.

---

### Decision · Program_Chairs · 2023-10-07

**Decision:**

Accept-Findings

**Comment:**

The paper investigates negation understanding in LMs. It proposes fine-tuning models on data that includes negation understanding and propose a two-part loss to balance fine-tuning with avoiding catastrophic forgetting. Experiments show that this recipe improves negation understanding for MLMs without hurting performance on other tasks.

The reviewers agree that the proposed recipe is interesting and experiments support the hypotheses well. There are some concerns regarding generalization to longer sequences and whether the proposed approach would generalize beyond the experiments in this work, both of which the authors acknowledge would be interesting future work.